# Risk of Symptomatic Intracranial Hemorrhage After Mechanical Thrombectomy in Randomized Clinical Trials: A Systematic Review and Meta-Analysis

**DOI:** 10.3390/brainsci15010063

**Published:** 2025-01-11

**Authors:** Abdullah Reda, Alireza Hasanzadeh, Sherief Ghozy, Hossein Sanjari Moghaddam, Tanin Adl Parvar, Mohsen Motevaselian, Ramanathan Kadirvel, David F. Kallmes, Alejandro Rabinstein

**Affiliations:** 1Department of Neurologic Surgery, Mayo Clinic, Rochester, MN 55902, USA; mohammed.abdullah@mayo.edu (A.R.); ghozy.sherief@mayo.edu (S.G.); kadir@mayo.edu (R.K.); 2Department of Radiology, Mayo Clinic, Rochester, MN 55902, USA; alirezahasanzadeh75@gmail.com (A.H.); g.h.sanjarimoghaddam@gmail.com (H.S.M.); taninadl74@gmail.com (T.A.P.); mohsen.motavaselian@gmail.com (M.M.); 3Department of Neurology, Mayo Clinic, Jacksonville, FL 32224, USA; rabinstein.alejandro@mayo.edu

**Keywords:** intracranial hemorrhage, thrombectomy, acute ischemic stroke, sICH, thrombolysis, randomized controlled trials

## Abstract

Background: Symptomatic intracranial hemorrhage (sICH) is the most dreaded complication after reperfusion therapy for acute ischemic stroke. We performed a meta-analysis of randomized controlled trials to estimate and compare risks of sICH after mechanical thrombectomy (MT) depending on the location of the large vessel occlusion, concomitant use of intravenous thrombolysis, timing of treatment, and core size. Methods: Randomized controlled trials were included, following a comprehensive search of different databases from inception to 1 March 2024. Random-effect models in a meta-analysis were employed to obtain the pooled risk ratios (RRs) and their corresponding 95% confidence intervals (95% CI) for sICH with MT, and were then compared to other reperfusion treatment regimens, including best medical treatment and intravenous thrombolysis (IVT). Results: MT in the anterior circulation was associated with a significantly higher risk of sICH as compared with no-MT (RR: 1.46; 95%CI: 1.03–2.07; *p* = 0.037). The risk of sICH was comparable between the MT and MT+IVT groups (RR: 0.77; 95%CI: 0.57–1.03; *p =* 0.079). There was no difference in sICH risk with MT as compared with no-MT within 6 h of last known well (RR: 1.14; 95%CI: 0.78–1.66; *p* = 0.485) and beyond that time (RR: 1.29; 95%CI: 0.80–2.08; *p* = 0.252); the risk of sICH was also comparable between MT conducted within 6 h of last known well and MT conducted beyond that time (*p* = 0.512). The sICH risk for MT in the posterior circulation (RR: 7.48; 95%CI: 2.27–24.61) was significantly higher than for MT in the anterior circulation (RR: 1.18; 95%CI: 0.90–1.56) (*p* = 0.003). MT was also associated with a significantly higher sICH risk than no-MT among patients with large core strokes (RR: 1.71; 95%CI: 1.09–2.66, *p* = 0.018). Conclusions: When evaluating cumulative evidence from randomized controlled trials, the risk of sICH is increased after MT compared with patients not treated with MT. Yet, the difference is largely driven by the greater risk of sICH in patients treated with MT for posterior circulation occlusions and, to a lesser degree, large core strokes. Concomitant use of intravenous thrombolysis and the use of MT in the extended therapeutic window do not raise the risk of sICH.

## 1. Introduction

Stroke is a sudden disruption in blood flow to the brain, leading to brain cell damage and potential long-term disabilities such as paralysis, speech problems, and cognitive impairment. It is a leading cause of death and disability worldwide, with around 15 million people affected each year. Key risk factors include high blood pressure, smoking, diabetes, obesity, and physical inactivity. Stroke outcomes can range from partial recovery to permanent disability or death [1,2,3]. Post-stroke care focuses on rehabilitation through physical, speech, and occupational therapy, along with medication to prevent recurrence and manage underlying health conditions. Reperfusion therapy (via mechanical thrombectomy (MT), intravenous thrombolytics (IVT), or a combination of both) is a successful treatment for acute ischemic stroke (AIS). Researchers usually differentiate between anterior and posterior circulation strokes in terms of management and prognosis due to differences in the characteristics and risks of MT and IVT in both populations. Prior research has conclusively demonstrated that MT improves outcomes in patients with large vessel occlusion (LVO) in the anterior circulation [4,5]. However, MT also increases the risk of symptomatic intracranial hemorrhage (sICH) [6,7] and sICH after reperfusion therapy raises the likelihood of poor functional outcomes and mortality [8]. Estimates show that sICH occurs in 2–14% of cases after MT [9,10,11,12]. Accurate identification of patients at high risk of sICH could assist in making treatment decisions and tailoring appropriate monitoring regimens after the intervention [13].

The efficacy of MT for basilar artery occlusions, which is a rare but serious stroke event resulting from an occlusion affecting the basilar artery, had been controversial until recent trials and a meta-analysis demonstrated better functional outcomes after MT versus medical management in selected patients [14]. Nevertheless, the sICH rate was higher among patients with basilar artery occlusion treated with MT than in the medical management group [14]. How the risk of sICH after MT for basilar artery occlusion compares with the MT for anterior circulation occlusion remains largely unknown.

Previous meta-analyses have compared the risk of sICH after MT alone versus medical management [15], or MT alone versus MT+IVT [16,17,18,19]. Yet no previous meta-analysis has comprehensively compared the risk of sICH across various cohorts of patients treated with MT, not just with and without IVT, but also anterior versus posterior circulation occlusions, extended time window, and large versus small core strokes. Therefore, we conducted a meta-analysis of randomized controlled trials to estimate the risks of sICH following MT compared to the best medical therapy (BMT, including IVT and other standard care practices and procedures) across these various patient groups.

## 2. Methods

### 2.1. Search and Screening

A systematic literature review of the English language literature was undertaken on 1 March 2024, according to the PRISMA standards for undertaking systematic reviews (although no protocol was established for conducting this review and, therefore, no registration number can be provided), and included a comprehensive search on PubMed, Embase, Web of Science, and Scopus from inception [20]. For each database, various combinations of potential keywords and/or MeSH phrases were employed to achieve this objective. The following keywords and MeSH phrases were included: “stroke”, “cerebral infarction”, “endovascular”, “thrombectomy”, “intracranial hemorrhage”, “sICH”, “ICH”, “hemorrhage”, and others. A full list of the exact search terms used for each database can be seen in Appendix A. In addition, we conducted a thorough manual check of the references cited in the included publications to ensure that no relevant papers were overlooked.

We included the randomized clinical trials (including post hoc analyses) that fulfilled the following criteria: (1) adult patients with acute ischemic stroke who had emergency treatment with MT, (2) sICH confirmed by subsequent imaging (no restrictions were made regarding the definition of sICH and the definitions of each trial will be presented as reported), and (3) comparative outcomes reported based on MT versus no-MT or MT versus MT+IVT. Concomitant anterior and posterior circulation strokes were included. Duplicates among the different databases were removed and the remaining articles were organized for screening according to the prespecified criteria. The abstracts were screened separately to identify the appropriate patient population and then thoroughly analyzed to determine if they met the criteria for inclusion. At least two reviewers also conducted full-text, blind evaluations of all possible studies, and final appropriateness for inclusion into the meta-analysis was confirmed by all authors.

### 2.2. Data Extraction and Outcomes

After performing a preliminary extraction, an extraction sheet was generated and the extraction was carried out by two authors. The extracted data comprised trial characteristics (name, year of publication, and country), stroke location, the definition of interventions of each population, age, gender, NIHSS and ASPECTS scores at baseline, the definition of sICH, the rates of good functional outcome at 90 days (mRS 0–2), 90-day mortality and successful recanalization, and door-to-puncture time. The main outcome of this analysis was the rates of sICH after MT. We also compared the risks of sICH for (1) anterior versus posterior circulation occlusions, (2) direct MT versus MT preceded by IVT, (3) conventional (≤6 h) versus extended (>6 h) therapeutic time window, and (4) core size large (ASPECTS < 6) versus not large (ASPECTS ≥ 6). Following the extraction process, a thorough revision of the retrieved data was conducted by a third author to ensure the absence of any previous errors.

### 2.3. Risk of Bias Assessment

The risk of bias in the included trials was assessed via the “Cochrane RoB 2: a revised tool for assessing risk of bias” [21]. Two reviewers evaluated all papers blindly. They assessed the quality of each study and resolved any disagreements with the help of a third author, if necessary.

### 2.4. Statistical Analysis

The package “meta” was utilized to perform a pairwise meta-analysis to obtain the pooled risk ratios (RRs) and their corresponding 95% confidence intervals (95% CI). The meta-analysis utilized a random-effects model due to the presence of methodological heterogeneity that contradicted the common effect assumption and because there were more than five trials included in the analysis. The pooled effect sizes were deemed heterogeneous if the *I*^2^ value exceeded 50% and/or the *p*-value was less than 0.05, as determined by the Q-statistic. A sensitivity analysis was also conducted by removing trials investigating large core occlusions to exclude any potential of increasing sICH risk.

## 3. Results

### 3.1. Search Results

The comprehensive search strategy adopted in this meta-analysis yielded 2851 citations, of which 542 duplicates were removed. Following this, 2195 articles were excluded after meeting the exclusion criteria when screened via title/abstract. Accordingly, 114 articles were screened for eligibility via a full-text assessment, which resulted in the final inclusion of 31 relevant trials that were eligible for data synthesis (Appendix A).

### 3.2. Characteristics and Quality Assessment

The data from 30 trials were extracted [11,22,23,24,25,26,27,28,29,30,31,32,33,34,35,36,37,38,39,40,41,42,43,44,45,46,47,48,49,50], in addition to the sICH rates data that were obtained from one unpublished trial (TESLA trial). These included 27 trials of anterior circulation stroke and 4 trials of posterior circulation stroke. Data on infarct core and reperfusion timing after stroke onset could be extracted from the 27 trials investigating anterior circulation strokes, including 6 that investigated large core occlusions and 9 with patients having reperfusion timing beyond 6 h from stroke onset (i.e., time of last known well). A further 8 trials compared MT versus MT+IVT, while 19 trials investigated MT versus no-MT (BMT only). The definition of sICH among the included trials was not consistent. The definition of sICH was parenchymal hemorrhage type 2 associated with an increase of ≥4 points in the NIHSS score or leading to death (SITS-MOST) in most of the included trials. The exact definition used in each trial is presented in Appendix A. Other characteristics of the included studies, including patients’ age, gender, baseline NIHSS, baseline ASPECTS, mRS score at 90 days, and other treatment outcomes are presented in Table 1. RoB assessment demonstrated that almost all trials had a low risk of bias, except for one trial that had a high risk of bias arising from the randomization process and another that had some concerns due to deviation from the intended intervention. The complete presentation of RoB assessment domains, excluding the one unpublished trial, is shown in Appendix A.

**Table 1 brainsci-15-00063-t001:** Characteristics of the included trials.

Trial	Country	Location	Design	Group 1											Group2									
				Total	Age	Male	Location	NIHSS	ASPECTS	mRS (0–2 90 Day)	90-Day Mortality	Successful Recanalization	Door to Puncture		Total	Age	Male	Location	NIHSS	ASPECTS	mRS	90-day Mortality	Successful Recanalization	Door to Puncture
**DIRECT-MT** [22]	China	A	RCT	327	69 [61–76]	189	ic-ICA: 112/320, M1: 161/320, M2: 42/320	17 [12–21]	9 [7–10]	119	58	243/306	84 [67–105]		329	18–60: 66; 60–80: 183; >80: 43	160	ic-ICA: 114/326, M1: 178/326, M2: 33/326	17 [14–22]	9 [7–10]	121	62	267/316	85.5 [70–115]
**RESILIENT** [23]	Brazil	A	RCT	11	65 [54–77]	60	left hemisphere: 64	18 [14–21]	8 [7–9]	39	27	91	170 [132–213]		111	67 [53–73]	57	63	18 [14–21]	8 [7–9]	22	33	-	161 [115–219]
**THERAPY** [24]	USA and Germany	A	RCT	43	67 (11)	27	left hemisphere: 33; ic-ICA: 18, M1: 31, M2: 6	17 [13–22]	7.5 [6–9]	19/50	12%	30/43	-		62	70 (10)	27	left hemisphere: 31; ic-ICA: 12, M1: 36, M2: 5	18 [ 14–22]	8 [7–9]	14/46	23.90%		102 [80–154]
**DAWN** [26]	USA	A	RCT	107	69.4 (14.1)	42	ic-ICA: 22, M1: 83, M2: 2	17 [13–21]	-	49%	20	90	109 [76–150]		99	70.7 (13.2)	51	ic-ICA: 19, M1: 77, M2: 3	17 [14–21]	-	13%	18	-	-
**RESCUE-Japan LIMIT** [25]	Japan	A	RCT	100	76.6 (10)	55	ic-ICA: 47, M1: 74, M2: 0; Tandem+M1: 20	22 [18–26]	3 [3–4]	14	18	86	254 [165–479]		102	75.7 (10.2)	58	ic-ICA: 49, M1: 70, M2: 3; Tandem+M1: 20	22 [17–26]	4 [3–4]	8	24	-	-
**RESCUE BT** [27]	China	A	RCT	483	67 [57–75]	294	ic-ICA: 98, M1: 310, M2: 77	16 [12–19]	8 [7–9]	219	84	439	398 [246–618]		462	68 [58–74]	263	ic-ICA: 96, M1: 305, M2: 62	16 [12–20]	8 [7–9]	228	82	427	400 [272–627]
**CHOICE** [28]	Spain	A	RCT	52	73 [69–67]	28	ic-ICA: 4, M1: 20, M2: 28	14 [10–20]	10 [8–10]	33	8	52	356 [260–635]		61	73 [71–76]	33	ic-ICA: 7, M1: 19, M2: 33	14 [8–20]	9 [9–10]	41	5	61	315 [218–680]
**SELECT2** [29]	United States, Canada, Europe, Australia, and New Zealand	A	RCT	178	66 [58–75]	107	ic-ICA: 80, M1: 91, M2: 7	19 [15–23]	4 [3–5]	4 [3–6]	68	142	109 [76–138]		174	67 [58–75]	100	ic-ICA: 66, M1: 100, M2: 8	19 [15–22]	4 [4–5]	5 [4–6]	71	-	-
**SWIFT PRIME** [30]	United States and Europe	A	RCT	98	65 (12.5)	54	ic-ICA: 17, M1: 62, M2: 13	17 [13–20]	9 [8–10]	59/98	9%	83%	224 [165–275]		97	66.3 (11.3)	45	ic-ICA: 15, M1: 72, M2: 6	17 [13–19]	9 [7–10]	33/93	12%	40%	-
**SKIP** [31]	Japan	A	RCT	101	74 [67–80]	56	ic-ICA: 41, M1: 19, M2: 41	19 [13–23]	7 [6–9]	60	8	91	-		103	76 [67–80]	72	ic-ICA: 36, M1: 18, M2: 49	17 [12–22]	8 [6–9]	59	9	96	-
**ATTENTION** [32]	China	P	RCT	226	66 (11.1)	149	VA-V4: 20, PBA: 69, MBA: 62, DBA: 74	24 [15–35]	9 [8–10]	75	83	208/223	5.6 [3.5–7.5]		114	67.3 (10.2)	82	VA-V4: 6, PBA: 39, MBA: 29, DBA: 40	24 [14–35]	10 [8–10]	12	63	-	-
**DEVT** [34]	China	A	RCT	115	70 [60–77]	66	ic-ICA: 18, M1: 95, M2: 3	16 [12–20]	8 [7–9]	63	20	113	200 [155–247]		117	70 [60–78]	66	ic-ICA: 17, M1: 99, M2: 2	16 [13–20]	8 [7–9]	55	21	117	210 [179–255]
**DIRECT-SAFE** [33]	Australia, New Zealand, China, and Vietnam	A	RCT	148	70 [61–78]	78	ic-ICA: 33, M1: 80, M2: 21, BA: 11, Tandem-ec: 27, ic-ASD: 6	15 [11–20]	10 [9–10]	80/146	22	127/ 143	87 [56–113]; n:145		147	69 [60–79]	88	ic-ICA: 31, M1: 83, M2: 23, BA: 8, Tandem-ec: 20, ic-ASD: 8	15 [10–20]	10 [9–10]	89/147	24	130/146	101 [75–127]; n:147
**TO-ACT** [35]	Netherlands, China, and Portugal	A	RCT	33	43 [33–50]	10	cerebral venous thrombosis	12 [7–20]	-	12 mo: 28/33	6 mo: 4/33	22 (79%)	-		34	38 [23–48]	7	cerebral venous thrombosis	12 [5–20]	-	12 mo: 28/34	6 mo: 1/33	15 (52%)	-
**Huu An, 2022** [36]	Vietnam	A	RCT	30	66.5 [59–78.5]	21	ICA: 33.3, M1: 60, M2: 6.7	12 [10–14]	7 [7–8]	60%	3.30%	90%	69.5 [51–84]		30	64 [58.75–74]	18	ICA: 40, M1: 50, M2: 10	13 [11–17.25]	7 [7–8]	60%	6.70%	86.70%	73.0 [63.25–86]
**THRACE** [11]	France	A	RCT	200	62.8 (13.0)	115	intracranial internal carotid artery, the M1 segment of the middle cerebral artery, or the superior third of the basilar artery	18 [15–21]	-	53%	12%	-	-		202	62.8 (14.4)	102	intracranial internal carotid artery, the M1 segment of the middle cerebral artery, or the superior third of the basilar artery	17 [13–21]	-	42.10%	13%	-	-
**DEFUSE 3** [37]	USA	A	RCT	92	70 [59–79]	46	ICA: 35%, MCA: 65%	16 [10–20]	8 [7–9]	3 [1–4]	13 (14)	65/83 (78)	0:59 (0:39–1:27)		90	71 [59–80]	44	ICA: 40%, MCA: 60%	16 [12–21]	8 [7–9]	4 [3–6]	23 (26)	14/77 (18)	-
**IMS 3** [38]	United States, Canada, Australia, and Europe	A	RCT	434	69 (23–89)	218	Left hemisphere: 224 (51.6), Right hemisphere: 197 (45.4), Brain stem or cerebellum: 10 (2.3), Unknown or multiple locations: 3 (0.7)	17 [7–40]	247 (56.9)	99.80%	19.10%	-	-		222	68 (23–84)	122	Left hemisphere: 106 (47.7), Right hemisphere: 109 (49.1), Brain stem or cerebellum: 4 (1.8), Unknown or multiple locations: 3 (1.4)	16 [8–30]	131 (59.0)	100	21.60%	-	-
**EXTEND-**IA [39]	Australia and New Zealand	A	RCT	35	68.6 (12.3)	17	ICA: 31%, MCA: 69%	17 [13–20]	-	1 [0–3]	-	94%	-		35	70.2 (11.8)	17	ICA: 31%, MCA: 69%	13 [9–19]	-	3 [1 to 5]	-	43%	-
**SYNTHESIS Expansion** [40]	Italy	A	RCT	181	66 (11)	106	Anterior circulation: 88%, Posterior circulation: 10%	13 [9–17]	-	-	-	-	-		181	67 (11)	103	Anterior circulation: 94%, Posterior circulation: 6%	13 [9–18]	-	-	-	-	-
**ESCAPE** [41]	Canada, USA, South Korea, Ireland, UK	A	RCT	165	71 [60.81]	79	ICA: 27.6, M1: 68.1, M2: 3.7	16 [13–20]	9 [8–10]	53%	10.40%	72.40%	-		150	70 [60–81]	71	ICA: 26.5, M1: 71.4, M2: 2.0	17 [12–20]	9 [8–10]	29.30%	19%	-	-
**REVASCAT** [42]	Spain	A	RCT	103	65.7 (11.3)	55	ICA: 25.5, M1: 64.7, M2: 9.8	17.0 [14.0–20.0]	7.0 [6.0–9.0]	43.7	-	-	269 [201–340]		103	67.2 (9.5)	54	ICA: 26.7, M1: 64.4, M2: 7.9	17.0 (12.0–19.0)	8.0 [6.0–9.0]	28.2	-	-	-
**BASICS** [43]	Netherlands, Brazil, Germany, France, Italy, Switzerland, Czech republic	P	RCT	154	66.8 (13.1)	100	-	21	-	35.1	38.30%	-	4.4 [3.3–6.2]		146	67.2 (11.9)	96	-	22	-	30.1	43.20%	-	-
**MR CLEAN–NO IV** [45]	Netherlands	A	RCT	273	72 [62–80]	161	ICA: 25, M1: 57.4, M2: 16.5	16 [10–20]	9 [8–10]	49.1	20.5	78.2	63 [50–78]		266	69 [61–77]	144	ICA: 18.8, M1: 65.4, M2: 15	16 [10–20]	9 [8–10]	51.1	15.8	84.7	64 [51–78]
**BAOCHE** [44]	China	P	RCT	102	64.2 (9.6)	80	Basilar-artery occlusion	20 [15–29]	8 [7–10]	39	31	-	153 [99–235]		88	63.7 (9.8)	79	Basilar-artery occlusion	19 [12–30]	8 [7–10]	14	42	-	-
**ANGEL-ASPECT** [46]	China	A	RCT	230	68 [61–73]	135	ICA: 36.1, M1: 63, M2: 0.9	16 [13–20]	3 [3–4]	30	21.70%	-	-		225	67 [59–73]	144	ICA: 36, M1: 63.1, M2: 0.9	15 [12–19]	3 [3–4]	11.6	20%	-	-
**MR RESCUE** [47]	USA	A	RCT	64	Penumbral: 66.4 (13.2) NonPenumbral: 61.6 (12)	30	ICA: Penumbral: 18 NonPenumbral: 23, M1: Penumbral: 53 NonPenumbral: 70, M2: Penumbral: 29 NonPenumbral: 7	Penumbral: 16 [12–18] NonPenumbral: 19 [17–22]	-	Penumbral: 21 NonPenumbral: 17	-	-	-		54	Penumbral: 65.8 (16.9) NonPenumbral: 69.4 (15.9)	27	ICA: Penumbral: 15 NonPenumbral: 10, M1: Penumbral: 68 NonPenumbral: 80, M2: Penumbral: 18 NonPenumbral: 10	Penumbral: 16 [11–18] NonPenumbral: 20.5 [17–23]	-	Penumbral: 26 NonPenumbral: 10	-	-	-
**TENSION** [48]	Canada and Europe	A	RCT	128	73 [65–81]	59	ICA: 41/125, M1: 83, M2: 0, MCA+ACA: 1, tandem: 8	19 [16–22]	-	4 [3–6]	40.00%	104	4.2 [3.4–5.9]		125	74 [64–80]	51	ICA: 37/127, M1: 88, M2: 1, MCA+ACA: 1, tandem: 7	18 [15–22]	-	6 [4–6]	51%	-	-
**BEST** [49]	China	P	RCT	66	62 (50−74)	48	Vertebral artery V4 segment: 7; Basilar artery: 59	32 (18–38)	8 (7–9)	22	22	45	114 (66–150)		65	68 (57−74)	52	Vertebral artery V4 segment: 5; Basilar artery: 60	26 (13–37)	8 (7–9)	18	25	9	111·5 (65–160)
**LASTE** [50]	France, Spain, and the United States	A	RCT	159	73 (66–79)	82	ICA: 69; M1 MCA: 88; Other: 2	21 (18–24)	2 (1–3)	4 (3–6)	57/158 (36.1)	48 (31.8)	66 (45 to 97)		165	74 (65–80)	88 (53.3)	ICA: 74; M1 MCA: 91	21 (18–24)	2 (1–3)	6 (4–6)	91/164 (55.5)	-	

A: Anterior circulation occlusions; P: posterior circulation occlusions; RCT: randomized controlled trial.

### 3.3. sICH Risk in the Anterior Circulation

#### 3.3.1. Effect of Treatment Type

MT was associated with a significantly higher sICH risk than no-MT (RR: 1.46; 95%CI: 1.03–2.07; *p* = 0.037), with no significant heterogeneity (*I*^2^ = 24%; *p* = 0.168). The risk was comparable between the MT versus MT+IVT groups (RR: 0.77; 95%CI: 0.57–1.03; *p =* 0.079), indicating that IVT administration with MT does not further increase the risk of sICH (Figure 1).

#### 3.3.2. Effect of Timing

Time from last known well did not affect the risk of sICH. As compared with no-MT, the risk of sICH after MT was not increased regardless of whether time from last known well was within 6 h (RR: 1.14; 95%CI: 0.78–1.66; *p* = 0.485) or beyond 6 h (RR: 1.29; 95%CI: 0.80–2.08; *p* = 0.252)]. The risk of sICH was also similar when comparing MT conducted within 6 h versus MT conducted after 6 h (*p* = 0.215) (Figure 2).

#### 3.3.3. Effect of Core Size

MT was associated with a significantly higher sICH risk than no-MT among patients with large core strokes (RR: 1.71; 95%CI: 1.09–2.66, *p* = 0.018), with no significant heterogeneity (*I*^2^ = 0%; *p* = 0.782). Conversely, in non-large core patients there was no difference in sICH risk between MT and no-MT groups (RR: 1.06; 95%CI: 0.76–1.49; *p* = 0.702), with no significant heterogeneity across studies (*I*^2^ = 33%; *p* = 0.072). However, when comparing sICH risks after MT for large core strokes versus MT for non-large core strokes, the difference was not significant (*p* = 0.089) (Figure 3).

### 3.4. Sensitivity Analysis

After the exclusion of large core trials, the sICH risk did not differ between patients treated with MT and no-MT (RR: 1.30; 95%CI: 0.75–2.24; *p* = 0.313) and between MT and MT+IVT (RR: 0.77; 95%CI: 0.57–1.03; *p* = 0.079) (Figure 4).

After the exclusion of large core trials, time from last known well did not affect the risk of sICH. Compared with no-MT, the risk of sICH after MT was not increased regardless of whether time from last known well was within 6 h (RR: 1.11; 95%CI: 0.74–1.67; *p* = 0.590) or beyond that time (RR: 0.88; 95%CI: 0.38–2.04; *p* = 0.691). The risk of sICH was also similar when comparing MT conducted within 6 h versus MT conducted after 6 h (*p* = 0.512) (Figure 5).

### 3.5. sICH in Anterior Versus Posterior Circulation Stroke

The risk of sICH was greater with MT than no-MT for posterior circulation occlusion (RR: 7.48; 95%CI: 2.27–24.61). No significant heterogeneity was noted (*I*^2^ = 0%; *p* = 0.955). On the other hand, the risk with MT was comparable to that of no-MT in patients with anterior circulation occlusions (RR: 1.18; 95%CI: 0.90–1.56), with no significant heterogeneity (*I*^2^ = 30%; *p* = 0.071). When comparing the sICH risk after MT for anterior versus posterior circulation occlusions, the risk was significantly higher among patients with posterior circulation occlusions (*p* = 0.003) (Figure 6). Sensitivity analysis removing the large core trials made this risk difference larger (RR: 7.48; 95%CI: 2.27–24.61 versus 1.06; 95%CI: 0.76–1.49; *p* = 0.002) (Figure 6).

## 4. Discussion

The current meta-analysis was aimed at summarizing the information on sICH from RCTs in various cohorts of patients treated with MT. MT has been associated with sICH as a potential adverse event despite the advantages correlated with recanalization and enhanced patient outcomes. The results show that the risk of sICH with MT is higher than no-MT, particularly for patients with posterior circulation occlusions, and to a lesser degree for patients with large core. Meanwhile, the risk of sICH is not increased when MT is combined with IVT or when MT is conducted after 6 h from last known well.

Data on anterior circulation strokes are much more extensive, allowing for adequate evaluation of sICH in subgroups of patients. Our analysis demonstrated that patients with anterior circulation stroke undergoing MT had a significantly higher risk of sICH than no-MT patients. Administration of IVT with MT does not increase the risk of sICH compared to MT alone, in agreement with previous meta-analyses [51,52], supporting the safety of bridge therapy for patients with anterior circulation large vessel occlusions [22,23,27,30,42]. Additionally, the timing of when MT is conducted does not have any major effect on sICH risk. Our meta-analysis showed that the risk of sICH was comparable when MT was conducted within the first 6 h after stroke onset versus in the extended therapeutic window.

We found that MT was associated with increased risk of sICH in patients with large core strokes, but not in patients with smaller core. In fact, the impact of core size on sICH risk may be underestimated. When interpreting the risk of sICH in patients with large core strokes, it is crucial to remember that even large parenchymal hemorrhages may not be categorized as symptomatic because of a ceiling effect on the NIHSS (i.e., patients with very high NIHSS strokes at baseline may not have a sufficient numerical increase in the NIHSS score for the ICH to be called symptomatic even if large and associated with major mass effects). This is a caveat to be considered when comparing the rates of sICH between strokes with large and non-large cores.

Our finding that the risk of sICH after MT is higher in patients with posterior circulation strokes as compared to strokes in the anterior circulation deserves a careful interpretation. Hemorrhage in posterior circulation strokes becomes symptomatic more often; even small hemorrhages in the brainstem or thalamus can cause major neurological decline. Meanwhile, cerebellar infarctions with hemorrhagic conversion from reperfusion injury can cause major deterioration from mass effect causing obstructive hydrocephalus or brainstem compression. Also, in practice, delayed diagnosis and treatment are more common in patients with ischemia affecting the posterior circulation. In such cases, MT may be attempted as a last resort intervention despite the presence of established infarction in areas where bleeding can be life-threatening. Also, most trials of posterior circulation stroke included patients with a medium-to-high severity only (NIHSS of 10 or more) [32,43,44], due to prior evidence suggesting that MT might not be beneficial in patients with mild posterior circulation strokes [43,53,54,55,56]. Further research to refine the prediction of sICH after MT for posterior circulation stroke would be useful, but the risk of sICH should not dissuade clinicians from pursuing MT in patients with severe deficits from a basilar artery occlusion.

The current meta-analysis has several strengths, particularly the evaluation of sICH risk related to MT in different patient groups and clinical scenarios. These multiple comparisons allowed us to identify groups of patients (basilar occlusion, large core, and others) with higher risk of sICH after MT. This meta-analysis also has limitations. sICH was reported in almost all trials as a secondary outcome without clear adjustments for potential confounders. The definitions of sICH among trials were not consistent. The number of trials investigating posterior-circulation stroke was relatively small, which restricted our ability to perform sub-analyses as we did for anterior circulation stroke (i.e., investigating the impact of treatment type, timing, and infarct core on risk of sICH). Furthermore, three of the four posterior circulation trials were conducted in China, which questions the generalizability of our findings to patients from other backgrounds.

## 5. Conclusions

Overall, the risk of sICH is increased after MT compared with patients not treated with MT, but the difference is largely driven by a greater risk of sICH in patients undergoing MT for basilar artery occlusions and in those presenting with large core strokes. The concomitant use of IVT and MT in the extended therapeutic window does not raise the risk of sICH. These findings shed light on the need to develop more personalized management practices for stroke patients, via enhancing the diagnostic efficacy and providing more precise recanalization approaches with a minimal risk associated with sICH development. Further research to refine the prediction of sICH after MT for posterior circulation stroke would be useful, but the risk of sICH should not dissuade clinicians from pursuing MT in patients with higher risks of sICH, as previously elaborated.

## Figures and Tables

**Figure 1 brainsci-15-00063-f001:**
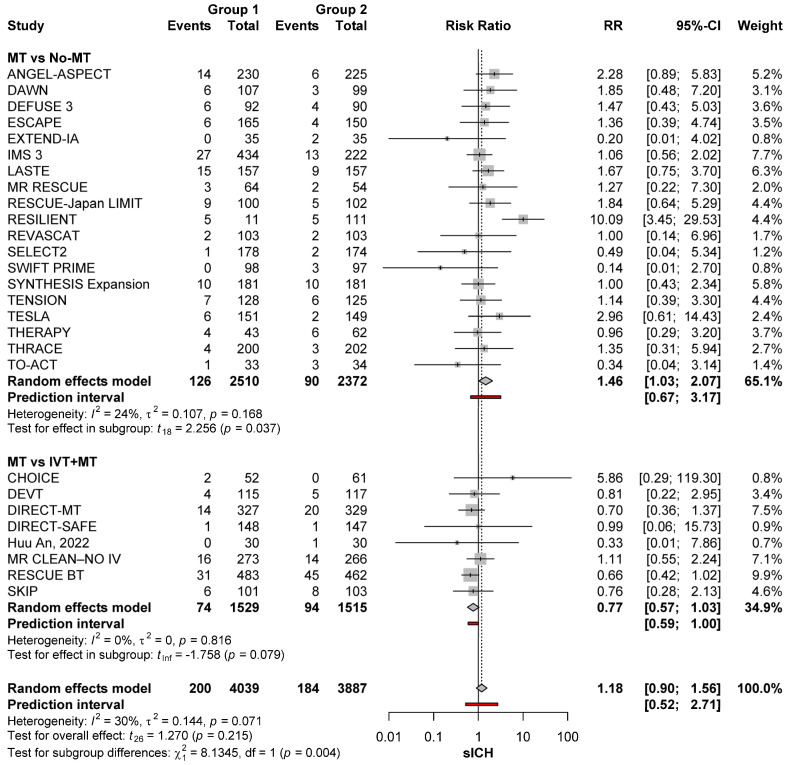
sICH risk with the different reperfusion regimens among patients with anterior circulation occlusions [11,22,23,24,25,26,27,28,29,30,31,32,33,34,35,36,37,38,39,40,41,42,43,44,45,46,47,48,49,50].

**Figure 2 brainsci-15-00063-f002:**
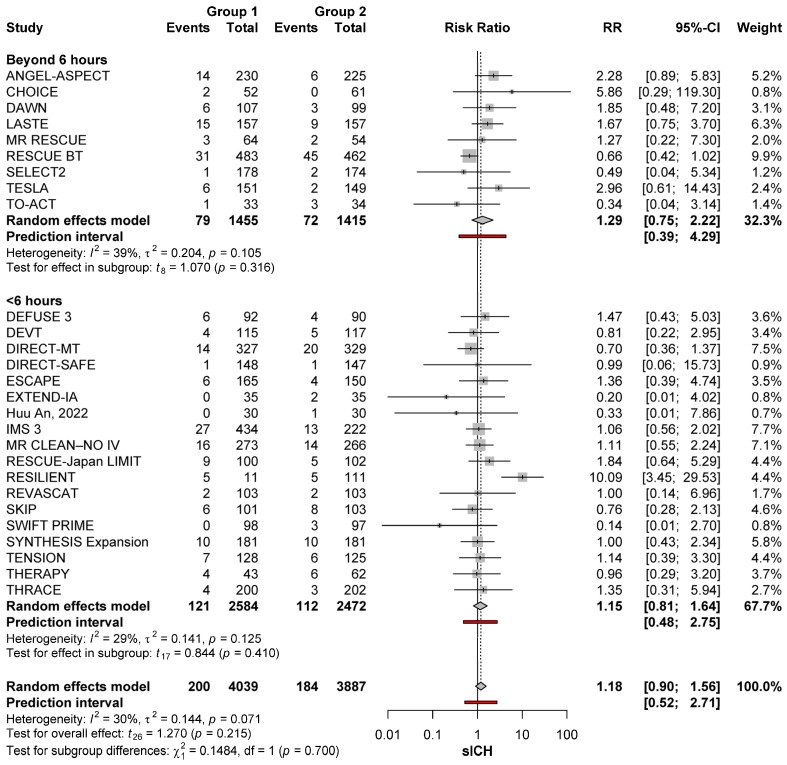
sICH risk with MT (group 1) and no-MT (group 2) based on timing from stroke onset in the anterior circulation [11,22,23,24,25,26,27,28,29,30,31,32,33,34,35,36,37,38,39,40,41,42,43,44,45,46,47,48,49,50].

**Figure 3 brainsci-15-00063-f003:**
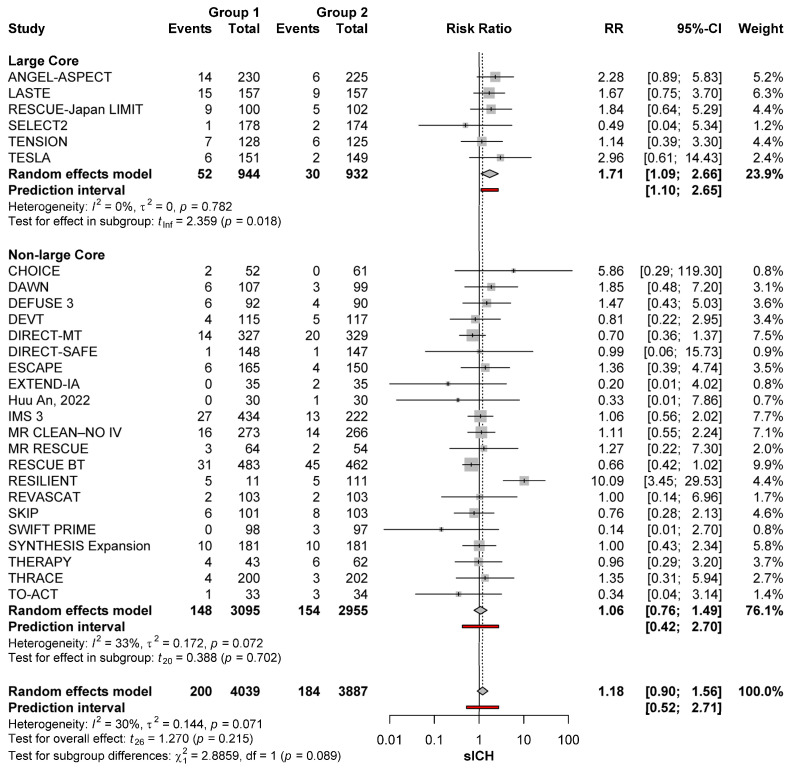
sICH risk with MT (group 1) and no-MT (group 2) in large and non-large core occlusions of the anterior circulation [11,22,23,24,25,26,27,28,29,30,31,32,33,34,35,36,37,38,39,40,41,42,43,44,45,46,47,48,49,50].

**Figure 4 brainsci-15-00063-f004:**
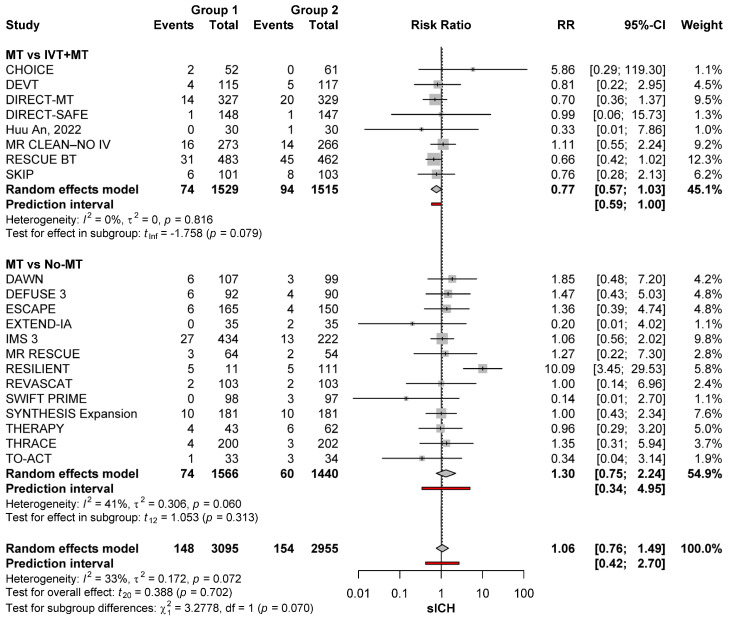
sICH risk with the different reperfusion regimens among patients with anterior circulation occlusions after removing large core occlusions [11,22,23,24,25,26,27,28,29,30,31,32,33,34,35,36,37,38,39,40,41,42,43,44,45,46,47,48,49,50].

**Figure 5 brainsci-15-00063-f005:**
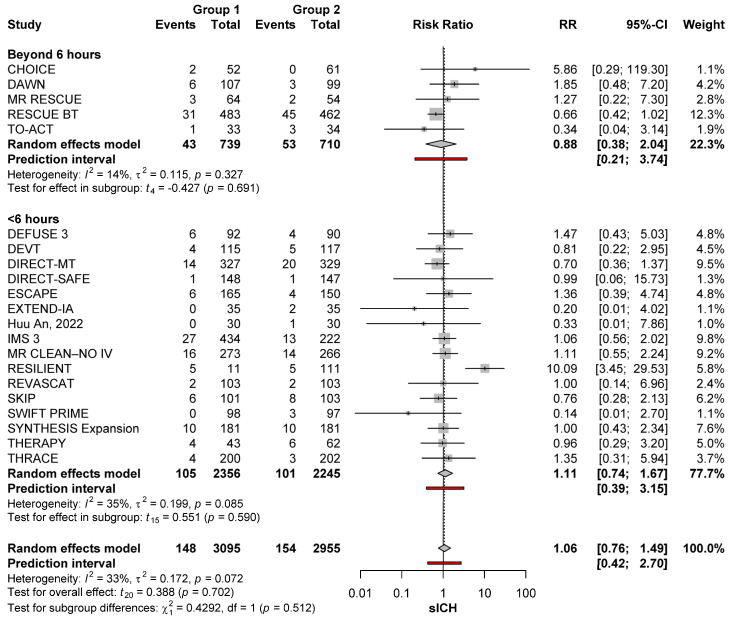
sICH risk with MT (group 1) and no-MT (group 2) based on timing from stroke onset in the anterior circulation after removing large core occlusions [11,22,23,24,25,26,27,28,29,30,31,32,33,34,35,36,37,38,39,40,41,42,43,44,45,46,47,48,49,50].

**Figure 6 brainsci-15-00063-f006:**
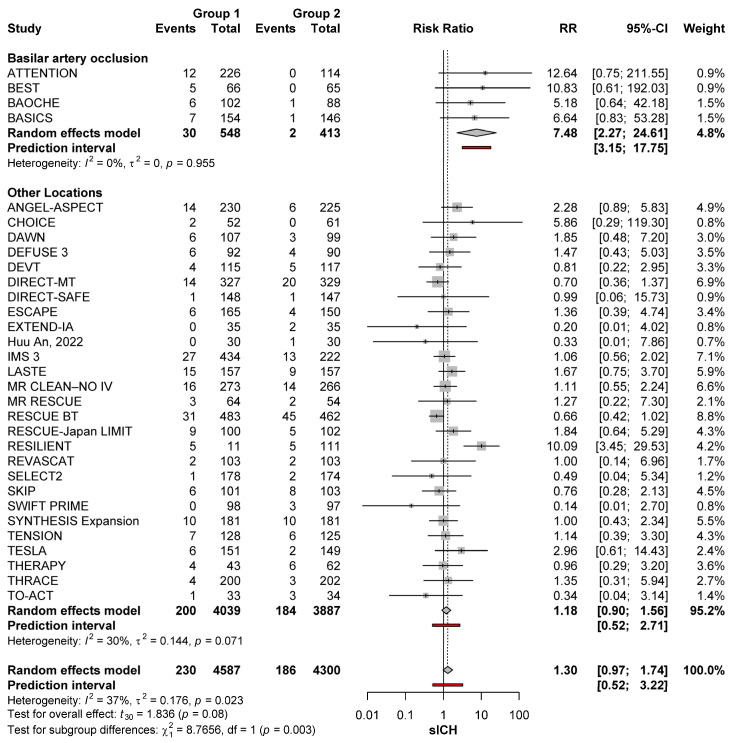
sICH risk with MT (group 1) and no-MT (group 2) in posterior occlusions and other locations [11,22,23,24,25,26,27,28,29,30,31,32,33,34,35,36,37,38,39,40,41,42,43,44,45,46,47,48,49,50].

## Data Availability

The data are not publicly available due to privacy.

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
