# Peer review of "Risk of Symptomatic Intracranial Hemorrhage After Mechanical Thrombectomy in Randomized Clinical Trials: A Systematic Review and Meta-Analysis"

_brainsci, 2025, doi:10.3390/brainsci15010063_

Round 1
Reviewer 1 Report
Comments and Suggestions for Authors
I have following comments:
1. The definition of symptomatic hemorrhage in Materials and Methods is given as a reference to supplemetary material. I think it would be easier for the reader to give at least a brief description of the definition of symptomatic hemorrhage in the text and then refer to the table in the suppl. mat.
2. The finding of an increased risk of bleeding with the otherwise positive effect of mechanical thrombectomy is somewhat surprising. Could you please indicate in the Discussion what are the possible causes or explanations, other than those mentioned in your higher risk subgroups.
3. Literature citations need to be consolidated and checked. E.g. missing journal titles. Introduce them according to the conventions of the journal.
4. Table 1 requires editorial revision. Possibly split into multiple tables.
Author Response
The authors would like to show gratitude to reviewer 1 for their efforts and time exerted in reviewing our manuscript. We tried our best to enhance the manuscript’s quality based on your suggestions, and we welcome any additional feedback. All changes have been highlighted using track-changes within the manuscript file.
- The definition of symptomatic hemorrhage in Materials and Methods is given as a reference to supplemetary material. I think it would be easier for the reader to give at least a brief description of the definition of symptomatic hemorrhage in the text and then refer to the table in the suppl. mat.
Response: Thanks for your comment. We agree with you and have briefed the definition of sICH in the text before referring to the exact definitions used by each trial in the supplementary materials. Changes were made via “track-changes”.
- The finding of an increased risk of bleeding with the otherwise positive effect of mechanical thrombectomy is somewhat surprising. Could you please indicate in the Discussion what are the possible causes or explanations, other than those mentioned in your higher risk subgroups.
Response: Thanks for your comment. It is well-established among all trials that MT is usually associated with sICH as a potential adverse event, which is logical because the procedure is considered invasive and carries a higher risk of complications that best medical treatment as elaborated in the discussion and among all included trials. We adjusted the manuscript using track changes.
- Literature citations need to be consolidated and checked. E.g. missing journal titles. Introduce them according to the conventions of the journal.
Response: Thank you. We agree with you. We have edited the citations.
- Table 1 requires editorial revision. Possibly split into multiple tables.
Response: We agree that the table is not fitting properly in the manuscript. However, we tried our best to make it concise and suitable within the manuscript file and removing any further information would not be the best option since all presented data are relevant. We believe that it can be enhanced further during production should the article be accepted for publication.
Reviewer 2 Report
Comments and Suggestions for Authors Manuscript ID: brainsci-3366188
Dear Authors,
As a result of my review, I recommend you focus on enhancing the methodology, clarity and applicability of the findings.
My comments:
1. Authors can adequately define "basilar artery occlusion" in the introduction.
2. Line no - 59: "Large core" refer clearly as it seems incomplete.
3. What did you consider BMT regarding Symptomatic intracranial hemorrhage (after MT). Please mention that too in the introduction.
4. Correct this sentence "These multiple comparisons allowed us to identified groups of patients"
5. Rephrase this: Concomitant use of intravenous thrombolysis and the use of MT in the extended therapeutic window do not raise the risk of sICH.
6. What are the impacts of regional differences and timing of treatments concerning results of your review?
7. Authors could add, "how do the findings contribute to current clinical practices in stroke management".
8. Supp. Figure 2. Risk of bias domains for all included trials: Labels and percentages were nor clear. Correct them.
9. Did you come across the impact of thrombectomy devices used in those procedures during your literature review?
10. Add future research directions in the conclusion section.
11. Briefly, mention about what is Stroke, its prevalence, risk factors, worst outcomes, ACS,and PCS in the introduction so that readers would get an overview of stroke.
Comments on the Quality of English Language
I recommend "language correction".
Author Response
Dear Authors,
As a result of my review, I recommend you focus on enhancing the methodology, clarity and applicability of the findings.
Response: The authors would like Reviewer 2 for their time and efforts exerted in reviewing our manuscript, which helped us enhance its quality. We did our best in replying to your concerns and enhancing the quality the different sections within our manuscript. All changes were highlighted with “track-changes”, and we welcome any additional feedback.
My comments:
- Authors can adequately define "basilar artery occlusion" in the introduction.
Response: Thanks for your comment. We added a definition in the introduction as suggested.
- Line no - 59: "Large core" refer clearly as it seems incomplete.
Response: Thanks for your comment. We edited the manuscript accordingly.
- What did you consider BMT regarding Symptomatic intracranial hemorrhage (after MT). Please mention that too in the introduction.
Response: Thanks for your comment. We agree with you. We added a description of the BMT in the introduction.
- Correct this sentence "These multiple comparisons allowed us to identified groups of patients"
Response: Thanks for your comment. We edited the sentence as suggested.
- Rephrase this: Concomitant use of intravenous thrombolysis and the use of MT in the extended therapeutic window do not raise the risk of sICH.
Response: Thanks for your comment. We edited the sentence as suggested.
- What are the impacts of regional differences and timing of treatments concerning results of your review?
Response: Thanks for your comment. The results of this meta-analysis show that the risk of sICH with MT is higher than no-MT, particularly for patients with posterior circulation occlusions, and to a lesser degree for patients with large core. Meanwhile, the risk of sICH is not increased when MT is combined with IVT or when MT is conducted after 6 hours from last known well. This means that further care should be taken for those patients to avoid or reduce the risk of sICH for these populations.
- Authors could add, "how do the findings contribute to current clinical practices in stroke management".
Response: Thanks for your comment. We discussed the implications of our findings in the discussion section and added more based on your suggestions in the conclusion section. We welcome additional comments.
- Figure 2. Risk of bias domains for all included trials: Labels and percentages were nor clear. Correct them.
Response: Thanks for your comment. We do not understand your point here. This figure shows the individual risk of bias domains assessment for all included trials, as low (green color), some concerns (yellow color), and high (red color) risk of bias, as elaborated below Supp. Figure 3.
- Did you come across the impact of thrombectomy devices used in those procedures during your literature review?
Response: Thanks for your comment. That’s a good point that was considered. However, doing such analysis required more specific data that could not be found in the included trials. Therefore, such analysis could not be conducted.
- Add future research directions in the conclusion section.
Response: Thanks for your comment. Future directions were provided across the discussion. We also elaborated more in the limitations and conclusions sections to make this more relevant.
- Briefly, mention about what is Stroke, its prevalence, risk factors, worst outcomes, ACS, and PCS in the introduction so that readers would get an overview of stroke.
Response: Thanks for your comment. We added to the introduction as suggested.
Comments on the Quality of English Language: I recommend "language correction".
Response: Thanks for your comment. We have carefully proofread the manuscript and edited the language flaws in our manuscript.
Round 2
Reviewer 1 Report
Comments and Suggestions for Authors
Please standardize the use of upper and lower case letters, including the use of abbreviations of scientific journals in references to the literature.
Otherwise, I have no further comments.
Reviewer 2 Report
Comments and Suggestions for Authors
Dear Authors,
You have revised your manuscript according to my suggestions. All the best.